# Recent Progress in Self-Powered Sensors Based on Triboelectric Nanogenerators

**DOI:** 10.3390/s21217129

**Published:** 2021-10-27

**Authors:** Junpeng Wu, Yang Zheng, Xiaoyi Li

**Affiliations:** School of Materials Science and Engineering, Ocean University of China, Qingdao 266100, China; wujunpeng@stu.ouc.edu.cn (J.W.); zhengyang@stu.ouc.edu.cn (Y.Z.)

**Keywords:** sensor, self-powered, TENG, contact electrification, application of TENG

## Abstract

The emergence of the Internet of Things (IoT) has subverted people’s lives, causing the rapid development of sensor technologies. However, traditional sensor energy sources, like batteries, suffer from the pollution problem and the limited lifetime for powering widely implemented electronics or sensors. Therefore, it is essential to obtain self-powered sensors integrated with renewable energy harvesters. The triboelectric nanogenerator (TENG), which can convert the surrounding mechanical energy into electrical energy based on the surface triboelectrification effect, was born of this background. This paper systematically introduces the working principle of the TENG-based self-powered sensor, including the triboelectrification effect, Maxwell’s displacement current, and quantitative analysis method. Meanwhile, this paper also reviews the recent application of TENG in different fields and summarizes the future development and current problems of TENG. We believe that there will be a rise of TENG-based self-powered sensors in the future.

## 1. Introduction

The first generators used Faraday’s principle of electromagnetic induction to convert other forms of energy into electricity, but large generators required a lot of materials and complicated equipment. Based on Maxwell’s shift currents, TENG collects energy from the surrounding environment and converts it into electricity [1]. The use of TENG reduces/replaces the dependence of various small electronic devices on traditional energy sources (such as chemical batteries) [2]. Compared with chemical batteries and traditional circuits, TENG greatly extends the service life of electronic products under long-term operation, reduces the maintenance times of miniaturized electronic devices, and minimizes environmental pollution due to its unique energy harvesting method. In addition, TENG does not need a lot of materials and large, expensive equipment, which greatly reduces the cost.

Sensors play an important role in developing the economy and promoting social progress [3,4,5]. As technology continues to evolve, many types of sensors have been developed [6,7,8]. The nanogenerator-based sensors with other harvesting principles (piezoelectric, pyroelectric, flexoelectric) have been studied extensively. TENG has unique harvesting methods of environmental energy, strong adaptability to the environment, and compatibility with small electronic devices [9]. In line with the trend of intelligent and miniaturized scientific and technological equipment in the current society, TENG has attracted more attention and more studies have been conducted on TENG [10,11]. The TENG of output powers, energies, voltages, and currents has been measured by Stanford research systems SR570 and an electrometer 6514 in usual conditions. From the theoretical study of triboelectrification to the use of TENG as a means of energy supply, to the static measurement of low-frequency signals using TENG voltage output in sensors, to the dynamic measurement of low-frequency signals using TENG current signals, the great potential of TENG development has been gradually explored.

In this paper, we discuss the principle and four main modes of TENG. It is confirmed that the TENG output can be improved by increasing the surface charge density *σ*, the effective contact area *S*, and the energy conversion efficiency by device design. Then, the progress of TENG-based self-powered sensors is systematically studied. Firstly, the application of TENG in the field of motion monitoring can monitor not only biological motion but also mechanical motion. There are also applications for collecting wave energy in the ocean using various models of TENG. As a self-powered sensor, TENG can monitor natural energy, toxic gases, and humidity conditions in the environment. TENG can also be used as an intelligent medical portable device for respiration, pulse monitoring, sterilization, and cardiac pacing systems. In the smart skin field, due to the high compatibility of TENG with portable devices, a TENG-based wearable sign language translation system and self-powered sensing friction skin intelligent soft actuator has emerged. The problem of capacitance and load matching in the TENG detection system and the cost of the TENG detector are discussed. TENG-based self-powered sensors show strong adaptability in the field of sustainability. After nearly a decade of development, it has flourished in several fields. It could solve some of the key problems facing the world’s sustainable development. This is also the goal of current and future TENG-based self-powered sensors. Therefore, the research progress of TENG-based self-powered sensor is summarized and prospected here. We believe that TENG-based self-powered sensors will develop more rapidly in the future and will soon be industrialized.

## 2. The Principle of TENG

### 2.1. Triboelectrification Effect (Contact Electrification)

With the progress of human society, people have a deeper and deeper understanding of static electricity and have gradually understood how static electricity is produced. One widely accepted theory is called triboelectricity. As the surfaces of two different materials contact each other, a limited amount of charge is transferred through the interface [2]. In general, the charge transfer occurs through the thermodynamic movement of electrons from a high-energy state occupied on one surface (valence band) to an unoccupied low-energy state (conduction band) on another surface. The contact potential difference (CPD) drives this movement. The energy distribution and state density of the two corresponding surfaces are proportional to CPD. The Fermi level of the conducting surface and the effective Fermi level of the insulator lead to CPD [12]. The energy gap between the acceptor and donor states is defined as the effective Fermi level of an insulator. As originally described by Duke et al. [13], in the case of conductor-insulator contact and insulation-insulator shell, any transferred charge to the insulating surface is confined and concentrated here instead of passing through immediately. The polarity of any charge remaining on the insulating surface after contact depends entirely on the direction of electron flow during the contact, which in turn depends on the polarity of the CPD driving the electron flow. This phenomenon of the accumulation of charge on the surface of the contact insulator is often referred to as contact charging. When two contact materials move horizontally across each other, the dynamic change of the contact charge is called triboelectricity or the triboelectric effect [14].

### 2.2. Triboelectric Sequence

After nearly a decade of research on TENG, the frictional electrification effect exists in the vast majority of materials contacted by human beings, thus the choice of TENG friction layers becomes diverse [15]. However, with different friction layers of material, the output signal is very different. It has been found that the ability of friction layer materials to gain and lose electrons is one of the main factors affecting TENG, which depends on their relative polarity. The triboelectric sequences of various common materials are determined by comparing relative polarity [16]. The more obvious the relative polarity difference between the two materials in the sequence, the more prominent the triboelectric effect of the two materials is, and the more the corresponding charge transfer amount is. These studies allow the generation of triboelectric sequences, in which materials are organized into a list in the order of the charges they generate on the surface. The materials that are sufficient to produce maximum CPD have been identified, and attempts have been made to create more quantitative triboelectric sequences against the corresponding materials. Because the structure relatively easily accommodates various functional groups, its dielectric properties and surface polarization are highly customized. A good understanding of transferred electron mechanisms has been established, including material parameters, the relationship between transferred charges, and environmental parameters, such as temperature and external electric fields. However, the more dynamic aspects of triboelectrification and the mechanisms remain to be fully understood.

### 2.3. Maxwell’s Displacement Current

In 2006, the concept of a nanogenerator (NG) was proposed by Zhong Lin Wang’s research group [17]. TENG was discovered in 2012, and more and more in-depth studies on NG have been conducted, and the theoretical system has become more and more perfect [1]. The most significant discovery was that Maxwell’s displacement current was the theoretical source of the nanogenerator [2]. The displacement field is denoted by *D*, the magnetic field is denoted by *B*, the electric field is denoted by *E,* the magnetization field is denoted *H*, the free charge density is denoted by $\rho_{f}$, the free current density is denoted by $J_{f}$, the polarization field density is *P*, and $\varepsilon_{0}$ represents the vacuum dielectric constant, Maxwell’s specific expression.

Gauss’s law describing how an electric charge produces an electric field:(1)$$\nabla \cdot D=\rho_{f}$$

Gaussian magnetic law without magnetic monopole:(2)∇·B=0

Maxwell–Ampere’s law:(3)$$\nabla \times E=-\frac{\partial B}{\partial t}$$

Faraday’s law:(4)$$\nabla \times H=J_{f}+\frac{\partial D}{\partial t}$$

The relation between *D* and *P* is:(5)$$D=\varepsilon_{0}E+P$$

For isotropic media, Equation (5) is defined as: $P=\left(\varepsilon -\varepsilon_{0}\right)E$, $D=\varepsilon_{0}E$. The second term in Equation (4) is defined as the Maxwell displacement current:(6)$$J_{D}=\frac{\partial D}{\partial t}=\varepsilon_{0}\frac{\partial E}{\partial t}+\frac{\partial P}{\partial t}$$

The displacement current is different from the current carried by free electrons that we normally observe but was due to the time-varying electric field (vacuum or medium) combined with the tiny time-varying motion of atom-bound charges and dielectric polarization in the material. In Equation (5), the first term and the second term were combined, and the displacement current becomes: $J_{D}=\varepsilon \frac{\partial E}{\partial t}$. In the triboelectric material, the displacement current has the polarization density caused by static charge:(7)$$J_{D}=\frac{\partial D}{\partial t}=\varepsilon \frac{\partial E}{\partial t}+\frac{\partial P_{s}}{\partial t}$$

The first term of displacement current is the theoretical basis of electromagnetic waves. The second is the basic theoretical basis and source of the nanogenerator, which shows that the current is caused by the static charge on the surface of the polarization field [18].

## 3. Main Working Mode and Quantitative Analysis Method of TENG

### 3.1. Main Working Modes of TENG

The concept of TENG was first proposed by Zhong Lin Wang’s research in 2012 [1]. After several years, more and more studies on TENG were conducted, and TENG has gradually been recognized and understood by everyone [18]. The collection and utilization of low-frequency energy in TENG has always been discussed. TENG has also been applied in various fields, such as sports monitoring, environmental monitoring, and ocean energy harvesting, as well as medicine and sports, demonstrating its adaptability to today’s society and technology. For so many types of TENG, there are four basic working modes.

The vertical separation mode TENG is shown in Figure 1a. In this structure, two layers of friction layer with different electronegativity are formed in the middle, and there are electrodes on the back of the friction layer. In the beginning when two layers of friction layer are in contact, due to the triboelectrification effect an equal number of opposite charges will be formed on the contact surface, and the positive and negative charges depend on the relative electronegativity of the friction layer. When the two friction layers begin to separate due to the principle of electrostatic induction, the electrode will be charged, and the inductive potential difference will be formed. When electrodes are connected, electrons move from one electrode to the other, and at the end of the separation the surface charge of the friction layer and the charge carried by the electrode reach an electrostatic balance. As the friction layers get closer and closer together, the electrons slowly flow backward until the friction layers are fully in contact, and the inductive potential difference between the electrodes is zero.

The horizontal sliding mode TENG is shown in Figure 1b. The initial structure is the same as that of the vertical friction mode TENG, which is charged by contact of the friction layer. Then, as the friction layers slide, opposite charges begin to be induced on the electrodes to maintain electrostatic balance, and an induced potential difference is formed. As the electrodes are connected, electrons migrate between the two electrodes. When the two friction layers are no longer in contact, the induced charge on the electrode and the friction charge on the friction layer reach an electrostatic balance and no electrons flow. After sliding, the two layers slowly contact each other, and the electrons slowly flow until the two layers are in full contact and the charge on the electrode is zero. This process is repeated, producing an electrical output of alternating current. The horizontal sliding configuration is not only suitable for plane sliding, but also cylinder sliding, disk sliding, and so on.

When substances of different electronegativity come into contact, they become charged. Some materials have frictional contact with a surface where the charge density reaches saturation, and the static charge can remain on the surface for some time. The simplified TENG structure of the sliding freestanding mode is shown in Figure 1c. The reciprocating movement of charged objects between two discontinuous electrodes through friction will cause a change in the potential difference, thus driving the continuous flow of free electrons between the electrodes.

The operating modes described earlier have two electrodes connected by a load. However, the single-electrode mode TENG (structure as shown in Figure 1d) has only one electrode, which is grounded or connected to a reference electrode. Contact with the charged friction layer by friction will change the charge distribution on the electrode when it is close to and away from the lower electrode according to the principle of electrostatic induction, so that the electric potential between the electrode and the reference electrode is constantly changing and free electrons flow between the two, forming a current.

This section is divided by subsections. They will provide a concise and precise description of the experimental results, their interpretation, and the experimental conclusions that can be drawn.

### 3.2. TENG Quantitative Analysis Method

The improvement of TENG’s output performance can be achieved in many ways, but output power, such as open-circuit voltage and short-circuit circulating current, is not measured by a unified standard. Different TENGs have different structures, number of electrodes, materials, effective contact area, maximum separation distance, mechanical frequency, load resistance, capacitance, and cost, which make it difficult to measure. At the same time, TENG is a configuration of pure capacitive devices, so the traditional method based on voltage-current (VI) diagrams to characterize most other energy harvesters—for instance, solar cells, thermoelectric generators, and electromagnetic generators—is not suitable for understanding the performance of TENG. We need to develop a set of unified standards to evaluate the performance of TENG and define its advantages. Wang et al. [19] designed a new method to understand the nature of the information generated by TENG, and the modified structural figure of merit (FOMCS) could be used to predict the charging characteristics of the TENG in the energy storage system, further promoting the development of TENG. As shown in Figure 2, Li et al. [20] designed the edge approximation-based equivalent capacitance method (EDAEC) for quantitative analysis of all modes of the TENG system, and this universal method is a milestone. It provides a more refined analysis model for the in-depth understanding of the working principles of different TENG systems.

## 4. Enhancing Output Performances

The main principle of TENG is the coupling of friction contact electrification effect and electrostatic induction effect. When the friction layer of TENG contacts, the contact surface of the friction layer has an equal and opposite charge quantity *Q* due to the friction contact electrification effect, which is equal to the product of the contact area S of the friction layer and the surface charge density *σ* on the friction layer. In the process of TENG friction layer separation, due to the principle of electrostatic balance, an equal and opposite amount of transferred charge *Qsc* is induced on the electrode of the friction layer. The higher the transfer charge *Qsc* is, the stronger the TENG output will be. Therefore, to improve the TENG output, the most important thing is to improve the transfer charge quantity *Qsc*.

At present, there are several ways to improve TENG output: increasing surface charge density σ, increasing effective contact area S, and improving energy conversion efficiency through equipment design. As shown in Figure 3a, Park et al. [21] proposed and designed a TENG with a friction layer consisting of polyvinylidene fluoride-trifluoroethylene (PVDF-TrFE)/MXene composite membrane and poly-ω-aminoundecanoyl (PA11) composite membrane. They doped MXene nanosheets into PVDF-TrFE composite films to form microscopic dipoles which increased the electronegativity of the friction layer and the surface charge density *σ*. Then electrostatic spinning technology was used to improve the surface volume ratio of nanofibers and to increase the effective contact area S of the friction layer. Due to the increasing of σ and S, the frictional charge *Q* increased, and thus the transferred charge *Qsc* and the TENG output increased.

Figure 3b shows that Sang-woo Kim et al. [22] designed a TENG with serrated electrodes. The main principle uses the gap between the sawtooth electrode and the wire to generate spark discharge, and the spark discharge allows a large number of electrons from the sawtooth electrode to move quickly to the wire through the device design to improve the energy conversion efficiency. The output voltage of TENG is 5 kV and the current density is 2 A m^−2^. The most common way is to change the surface structure of the friction layer to obtain a larger effective contact area, obtaining a larger amount of friction charge *Q*. As can be seen in insets of Figure 3c, Xu et al. [23] reported a TENG based on nickel and copper double-metal hydroxyl-nano wrinkles and nanometer nickel-copper double-metal hydroxide by folding the scale of the microsurface topography, greatly increasing the area of effective contact frictional contact *S*. TENG output also increased dramatically, and the output of the same friction layer of TENG increased 13 times. Similarly, Bae et al. [24] also formed micro-domes and nano-pores on the TENG friction layer to realize nano/microstructure on the friction electric surface, increasing its total surface area by 210% and improving the TENG output, as presented in Figure 3d. Figure 3e shows that Wang et al. [25] prepared a kind of TENG for electret film. Due to the charge injection principle of electret film, the surface charge density σ of the friction layer greatly increased, and the short-circuit current and open-circuit voltage could achieve up to about seven times that of traditional TENG. However, much work has been done to improve energy conversion efficiency through friction surface optimization, ion implantation, or device design, but none of these methods can break through the fundamental limitations of the inevitable electrical breakdown effect, which limits the output energy density. Illustrated in Figure 3f, Zi et al. [26] proposed a high-pressure gas environment to suppress the breakdown effect. Increasing the gas pressure between the friction layers to prevent gas breakdown can increase the maximum TENG energy density by more than 25 times in the contact separation mode and by more than five times in the sliding independent mode.

## 5. Application of TENG in the Field of Motion Detection

TENG has great application potential in low-frequency energy collection. TENG, as a self-powered sensor, could collect and detect various environmental energies in human living activities and nature [27]. TENG could get rid of the limitation of traditional batteries, greatly improving the portability and usability of the sensor [28].

For example, as presented in Figure 4a, Lee et al. [29] prepared a cotton sock-style TENG with a sensitivity of 0.06 V·N^−1^, which realized low-frequency movement energy collection and monitoring of various physiological signals (e.g., gait, contact force, sweating level, etc.), providing a way of thinking for portable intelligent devices in the future. As a self-powered sensor, TENG also has great application potential in the field of biological motion monitoring. Figure 4b shows that the TENG-based self-powered pressure sensor prepared by Wei et al. [30] can monitor the motion of various joints during movements, such as the motion of hands, elbows, armpits, knees, feet, and other different motion speeds. People breathe differently (at different rates and depths) at different levels of exercise. Zheng et al. [31] prepared a machine washable and stretchable TENG that can record human respiration rate and depth, which is very promising for further application in the field of sports, as shown in Figure 4c. In table tennis, players adjust and change their state by hitting position and hitting power. Wang et al. [32] prepared a TENG with a square grid structure which is used to collect vibration energy and sense pulse force to accurately display the position of each shot and impact force so that athletes can monitor their state in real-time and make adjustments and changes quickly, which can be seen in Figure 4d. TENG, as a sensor, can monitor not only human movement but also mechanical movement. Wang designed a self-powered TENG which can monitor wheel temperature and wheel speed in real-time during train running [33]. As presented in Figure 4e, Chen et al. [34] developed a self-powered motion tracking system to monitor the speed, direction, acceleration, start and end positions, and even the motion path of moving objects. This research will provide a new idea for TENG in the field of self-powered motion detection. Figure 4f shows that Wang et al. [35] also designed a novel sweep-type TENG which can directly monitor drivers’ driving habits and reflect road conditions through the output of TENG and has great application potential in future intelligent transportation.

## 6. Application of TENG-Based Sensors in Marine Fields

In total, 70% of the earth’s surface is ocean, which contains a lot of energy. Ocean energy is pollution-free and recyclable, known as blue energy. TENG has low dependence on the environment and can collect disordered low-frequency marine energy, which has attracted more and more attention [36]. As presented in Figure 5a, TENG, a tower-based device, can efficiently convert wave energy in any direction to electrical energy, providing a feasible method for collecting ocean energy on a large scale [37]. Figure 5b shows that Wang et al. [38] combined and utilized the spring-assisted structure and swinging structure TENG. A TENG of this structure can transform low-frequency water wave vibration into high-frequency motion, thus increasing the frequency of current output by 205 times, which is also a method of rapid and large-scale collection of ocean energy. The contact separation mode is generally adopted for marine energy collection TENGs, but the material will be abrased during the process of TENG operation and its output performance will be reduced, while the surface friction charge of non-contact suspended TENG will gradually decay without charge replenishment. However, Figure 5c details Wang et al.’s [39] adoption of a flexible rabbit hairbrush and segmental pair structure for swinging TENG, which effectively extended the energy collection time, reduced abrasion, and extended the service life of TENG, thus improved the total energy conversion efficiency. Most of the TENG models used to collect ocean energy are solid-solid models, but some of the energy is lost in the form of heat energy during the operation of solid-solid TENG. Solid-liquid TENG can greatly reduce the energy loss and mechanical loss in the friction process. In Figure 5d, Wang et al. [40] designed a simple open-structure TENG that can maintain output in a variety of marine environments and weather conditions, providing an efficient way to obtain all-weather marine energy in a real marine environment. In the ocean, providing sustainable and cost-effective marine distributed buoys has been a challenge. However, Figure 5e details how Xu et al. [41] designed a sandwich structure TENG that can continuously light a high brightness LED, providing an effective solution to solve the distributed buoy in the ocean problem. TENG has been gaining more and more attention in the ocean as an energy collector and sensor. However, most TENG become damaged or lost in the ocean, which can affect the marine environment. Wang et al. [42] prepared a seawater-degradable TENG which can convert wave vibration energy into electricity without maintenance and without damaging the environment. In the insets of Figure 5f, the first example of combining solar-driven interface evaporation with water wave detection is demonstrated. Self-powered sensors in the combined TENG and solar fields have great application potential [43]. At the same time, the device can detect several water wave parameters (frequency, height, velocity, and wavelength) in real-time by integrating the brake steam generator with TENG. It provides a feasible idea for clean water production in open water and for engineering self-powered water wave detection, prediction, and blue energy acquisition systems [44].

## 7. Application of TENG in Environmental Monitoring

Nature contains a variety of energy, such as wind, light, and water energy, and humans are increasingly experienced in collecting and monitoring this energy. As technology advances, electronics get smaller and smaller, and batteries become more and more troublesome to replace. TENG, as a highly efficient self-powered sensor, blurs the boundary between traditional batteries and sensors and improves energy efficiency. In the natural environment, the instability of TENG energy acquisition and output restricts its application and development. As shown in Figure 6a, Yu et al. [45] proposed gravity TENG, which can convert wind energy into stable electric energy and can realize the stable collection of natural wind energy. Poisonous gas in the living environment can seriously damage human health, so the poisonous gas monitoring device with an early warning function is of great significance. As presented in Figure 6b, the TENG prepared by Feng et al. [46] combines a metal oxide semiconductor (MOS) with a highly integrated self-powered unit, thus achieving accurate detection of toxic gases from aniline. Recycling waste energy-harvesting materials can not only reduce environmental pollution but also recycle environmental energy to generate renewable electricity and power wireless electronic equipment. Therefore, our lives will be more environmentally friendly. Figure 6c shows that a TENG prepared by Wang et al. [47] from wasted milk cartons can be used for pH detection and early landslide monitoring in the natural environment. The wind information is converted directly into an inductive electrical signal by TENG’s active wind sensor, but the signal processing and transmission of this active wind sensor still require an external power source. As shown in the insets of Figure 6d, Wang et al. [48] prepared the breeze-wind-driven TENG which is used to harvest wind energy and sense wind speed, realizing a complete set of the self-powered intelligent wireless sensing systems. Most TENG cannot work well in harsh environments (strong acids and bases), but Long et al. designed an acid-base-resistant TENG that can work well in harsh environments, providing the basis for a self-powered sensor for TENG to work in harsh environments, as presented in Figure 6e. The output performance of most TENGs is affected by ambient humidity [49] As humidity can change the charging phenomenon at the contact interface, the surface charge density dissipates rapidly. As shown in Figure 6f, Wang et al. [50] demonstrated a TENG with a kind of biofilm material which can achieve high output in high humidity, providing ideas for early warning devices in oceans and other high humidity environments.

## 8. Medical Applications of TENG Sensors

As portable devices become more and more popular, self-powered portable intelligent medical devices are attracting more and more attention. TENG has long been proven to be a reliable self-powered sensor and energy collector. As shown in Figure 7a, Hu et al. [51] prepared a low-cost, ultra-sensitive, self-powered pressure sensor with 150 mV·Pa^−1^. The ultra-sensitive pressure sensor can be installed on the chest and wrist of the human body to monitor respiration and pulse, respectively, which has a great application prospect in the field of intelligent medical equipment. TENG can be a sensor, but most of them use metal electrodes as metal has a certain hardness and is extremely impermeable, affecting wearability. As presented in Figure 7b, Qiu et al. [52] demonstrated a type of TENG with polyaniline (PANI) electrodes that can monitor patients’ respiratory status in real-time and give an alarm when they stop breathing, which has great application potential in the field of critical monitoring. There are many techniques for rehabilitation after physical injury, but most of the rehabilitation equipment requires an external power source. However, Bhatia et al. [53] prepared a portable TENG for patient rehabilitation, which has the potential to be a practical tool for patient rehabilitation and can be seen in the insets of Figure 7c. It is particularly dangerous for patients to be left alone during an infusion and at the end of infusion. Therefore, Yang et al. [54] prepared a liquid-solid contact super-hydrophobic TENG, as shown in Figure 7d, which can realize intelligent blood transfusion monitoring and real-time blood transfusion monitoring. Skin wounds are common in everyday life, but bacterial infections can seriously affect healing. Figure 7e shows that Tao et al. [55] proposed a flexible TENG patch which can kill nearly 100% of escherichia coli and Staphylococcus aureus and can greatly promote the proliferation and migration of fibroblasts, providing a convenient solution for the treatment of infected wounds. In smart medical devices, self-powered implants can extend the operating time of the device in the body and reduce the need for high-risk repeat surgery. In Figure 7f, Kim et al. prepared a commercial coin-sized TENG with a high-performance inertial drive based on body motion and gravity and realized a self-charging cardiac pacemaker system, which has great application prospect in the field of implantable electronic devices [56].

## 9. Application of TENG in the Smart Skin Field

With the popularity of smart skin, TENG came into view due to its superior biocompatibility and portability. More and more research has been conducted on TENG in the smart skin field [57,58]. As shown in Figure 8a, Cao et al. [59] demonstrated a TENG for intelligent electronic skin using a fish bladder membrane as friction. This TENG has good biocompatibility and good sensitivity of about 446 nA·s^2^ m^−1^ and 50 nA%^−1^ RH, which can monitor the position of charged objects within a range of 0–27 mm from the device. Most TENG as smart skin, in the process of working with the increase of time, will produce varying degrees of damage, affecting TENG performance. As presented in Figure 8b, Wu et al. designed a TENG that can be fully self-healing under environmental conditions [60]. TENG with a transmissibility of 88% can be stretched to more than nine times its original length and can be used as a self-powered active tactile electronic skin. TENG has high transmittance (88%) and good elasticity (up to nine times its original length and can be used as a self-powered active tactile electronic skin. The connection system of TENG and a dielectric elastomer actuator (DEA) was the application prospect of triboelectric nanogenerator in electronic skin and soft robots. Figure 8c shows that Wang et al. proposed a TENG-tunable intelligent light modulator that can facilitate the practical research of TENG-DEA systems in the field of microcomputer systems and human-computer interaction [61]. 

TENG, as a smart skin, has irreplaceable portability and comfort. A TENG-based self-powered sensor has great potential in sign language interpretation. Park et al. [62] presented a human skin-inspired, highly sensitive, sustainable, self-powered triboelectric flex sensor (STFS) for sensing the human finger gestures and converting the correlated sign language into voice and text. Lee et al. [63] showed a sign language recognition and communication system comprising triboelectric sensor integrated gloves, AI block, and the VR interaction interface.

In the insets of Figure 8d, Chen et al. demonstrated a TENG-based wearable sign language translation system which can accurately translate American Sign Language gestures into speech. The work indicated that the recognition rate was as high as 98.63% and the recognition time was less than 1 s [64]. In the development process of the smart skin field, various sensors are inevitably integrated with robots. Cao et al. demonstrated a sensing friction skin-intelligent soft actuator which can be used to evaluate grasping, sensing, and energy collection performance [65]. For smart skin, ultra-high sensitivity, biocompatibility, and transparency are very important indicators. You et al. designed TENG-based soft and stretchable smart skin. This smart skin can withstand up to 600% pressure and light transmittance up to 62.5% [66]. The ability to provide sustainable energy for personal electronic products provides an effective method for self-powered portable electronic devices to achieve sustainable operation. Figure 8e details how the integration of wireless technology and stretch electronics is critical for human–computer interaction, and Wang et al. demonstrated a highly stretchable transparent wireless electronic device consisting of silver nanofiber coils and functional electronics for power transmission and information communication, further advancing TENG’s development in the field of electronic skin [67]. As shown in Figure 8f, Wang et al. [68] demonstrated a TENG-based tactile sensor that is highly stretchable and transparent and can detect objects made from any common material. This TENG has great potential for tactile sensing and touchpad technology applications.

## 10. The Matching of Capacitance and Load in TENG Detection Systems and the Cost of TENG Detectors

TENG, to study electrical output, is often regarded as a capacitance model. When TENG is connected to external load storage, the power output can be low owing to system mismatches or incomplete charge transfer. Standardization of TENG takes into account environment-related life cycle assessment (LCA) and power equalization cost (LCOE), including cost, labour, recovery process, and so on [69]. A lot of research results have been obtained in the environmental life cycle evaluation of collection technologies. However, there is little research on the LCA of TENG, while research on the LCA of other energy collectors is ongoing. LCA focuses on environmental hotspots and global environmental impacts. Therefore, environmental profile and cost analysis for environmental life cycle evaluation and technical-economic analysis (TEA) is of great importance. Assessing the environmental status and carbon footprint of TENG will help reduce the cost of energy production for TENG and provide indicators of whether TENG poses new challenges to the environment.

## 11. Summary and Perspective

TENGs have attracted more and more attention as efficient low-frequency energy collectors and sensors. In this paper, the TENG-based self-powered sensors, including the triboelectrification effect, Maxwell’s displacement current, and quantitative analysis method, were systematically introduced. We reviewed recent advances in self-powered TENG-based sensors (such as sports, environment, medical, and other fields). We discussed the factors of TENG-based sensor output: surface charge density σ, effective contact area S, and energy conversion efficiency through equipment design. However, as a self-powered sensor, how to further improve the sensitivity of TENG, how to solve the mechanical loss of its friction, and how to improve its durability are also problems to be solved. We must consider the TENG quantitative analysis method and the cost of the TENG detector during its industrialization. We believe that TENG-based sensors will flourish in the future through technical perfection. The TENG-based self-powered sensor is a feasible way to realize sustainable self-sustaining micro/nanosystems in nanotechnology. It has great potential for applications in sensing, medicine, infrastructure/environmental monitoring, defense technology, and even personal electronics. Self-powered sensors can solve some of the key problems facing the world’s sustainable development. This is the goal of TENG-based self-powered sensors, now and in the future.

## Figures and Tables

**Figure 1 sensors-21-07129-f001:**
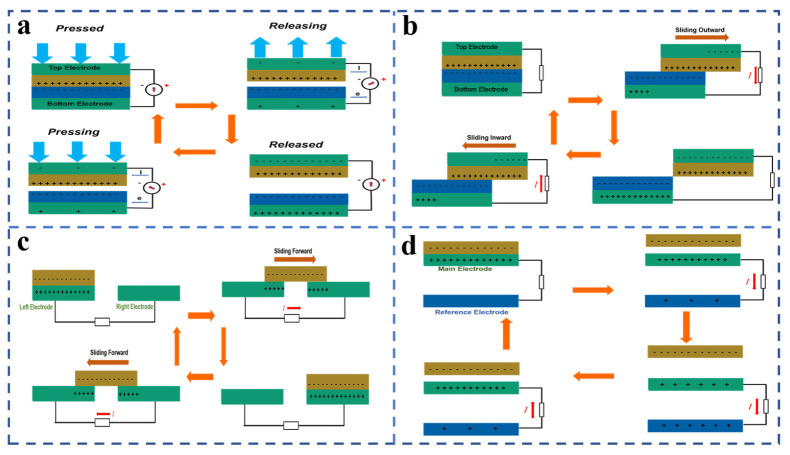
(**a**) TENG in vertical separation mode. (**b**) TENG in horizontal sliding mode. (**c**) TENG in sliding freestanding mode. (**d**) TENG in single–electrode mode.

**Figure 2 sensors-21-07129-f002:**
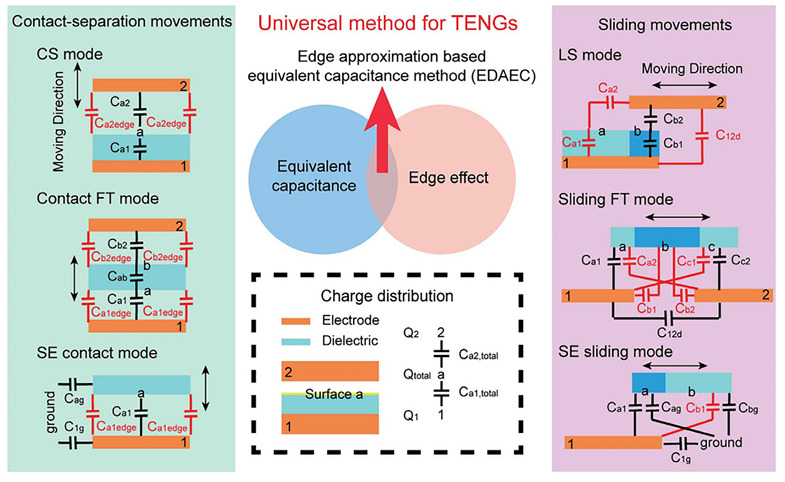
Schematic diagram of general methods for different types of TENG (Reproduced with permission [20]. Copyright 2019 Royal Society of Chemistry).

**Figure 3 sensors-21-07129-f003:**
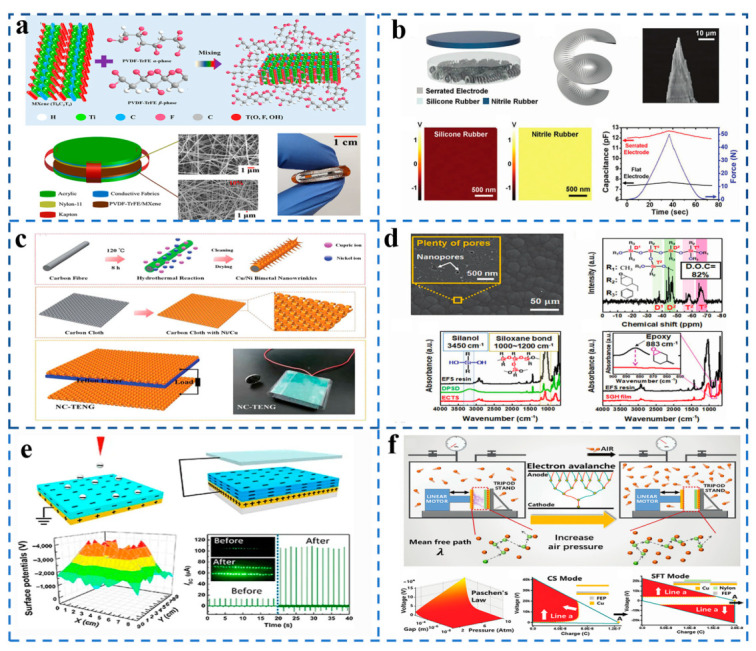
The method of TENG output performance improvement. (**a**) The electronegativity of the friction layer improves by doping with MXene nanosheets. (Reproduced with permission [21]. Copyright 2021 Elsevier). (**b**) The TENG improves energy conversion efficiency through serrated electrodes. (Reproduced with permission [22]. Copyright 2020 Elsevier). (**c**) The nanowrinkle structure of the friction layer increases the effective contact area of TENG. (Reproduced with permission [23]. Copyright 2020 Elsevier). (**d**) The layered surface nano/microstructure on the triboelectric surface increases the effective contact area of TENG. (Reproduced with permission [24]. Copyright 2020 American Chemical Society). (**e**) The electret charge injection principle greatly increases the surface charge density of the TENG friction layer. (Reproduced with permission [25]. Copyright 2016 Elsevier). (**f**) Using a high-pressure gas environment to suppress the breakdown effect increases the TENG output. (Reproduced with permission [26]. Copyright 2021 Springer Nature).

**Figure 4 sensors-21-07129-f004:**
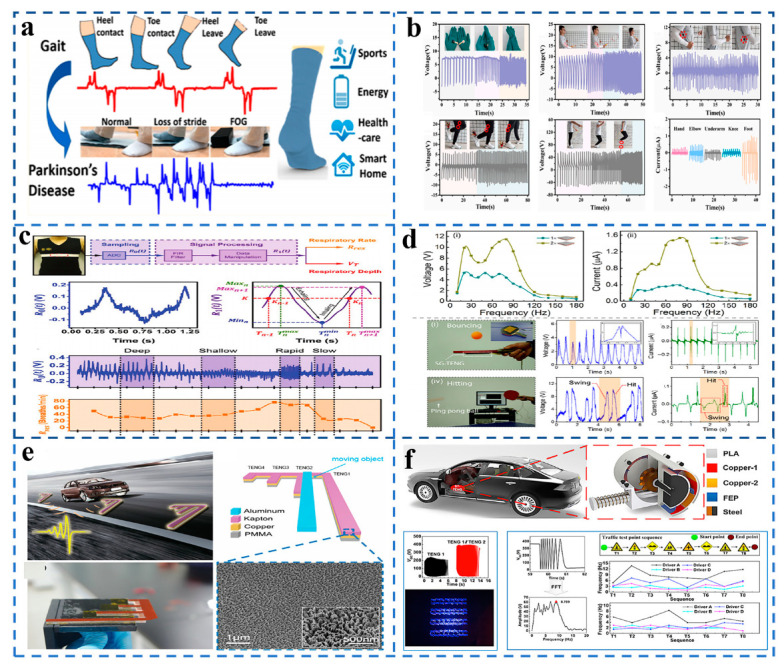
(**a**) Self-powered and self-functional cotton sock-style TENG. (Reproduced with permission [29]. Copyright 2019 American Chemical Society). (**b**) Self-powered pressure sensor by biomass-based, wearable TENG. (Reproduced with permission [30]. Copyright 2021 Elsevier). (**c**) TENG can monitor human respiratory information, including rate and depth. (Reproduced with permission [31]. Copyright 2016 John Wiley and Sons). (**d**) Square-grid TENG for harvesting vibrational energy and sensing impulsive forces. (Reproduced with permission [32]. Copyright 2017 Springer Nature). (**e**) TENG can be used for wheel safety monitoring. (Reproduced with permission [34]. Copyright 2021 John Wiley and Sons). (**f**) A sweep-type TENG can be used to monitor a driver’s habits. (Reproduced with permission [35]. Copyright 2020 Elsevier).

**Figure 5 sensors-21-07129-f005:**
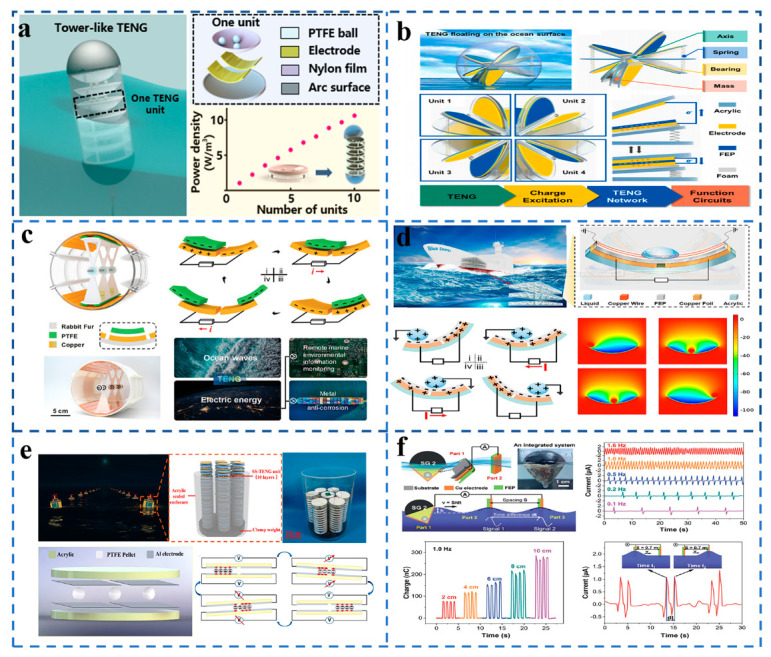
(**a**) TENG based on a tower-like structure is used to collect wave energy from any direction. (Reproduced with permission [37]. Copyright 2019 American Chemical Society). (**b**) Spherical TENG based on spring-assisted swing structure harvests water wave energy. (Reproduced with permission [38]. Copyright 2021 Elsevier). (**c**) TENG made of a flexible rabbit hairbrush and segmented structure. (Reproduced with permission [39]. Copyright 2021 John Wiley and Sons). (**d**) TENG based on droplets can be used to collect ocean wave energy. (Reproduced with permission [40]. Copyright 2021 American Chemical Society). (**e**) TENG that can be used for self-powered navigation buoys. (Reproduced with permission [41]. Copyright 2021 Elsevier). (**f**) A schematic diagram and a digital image of integrated steam generator and TENG. (Reproduced with permission [43]. Copyright 2020 John Wiley and Sons).

**Figure 6 sensors-21-07129-f006:**
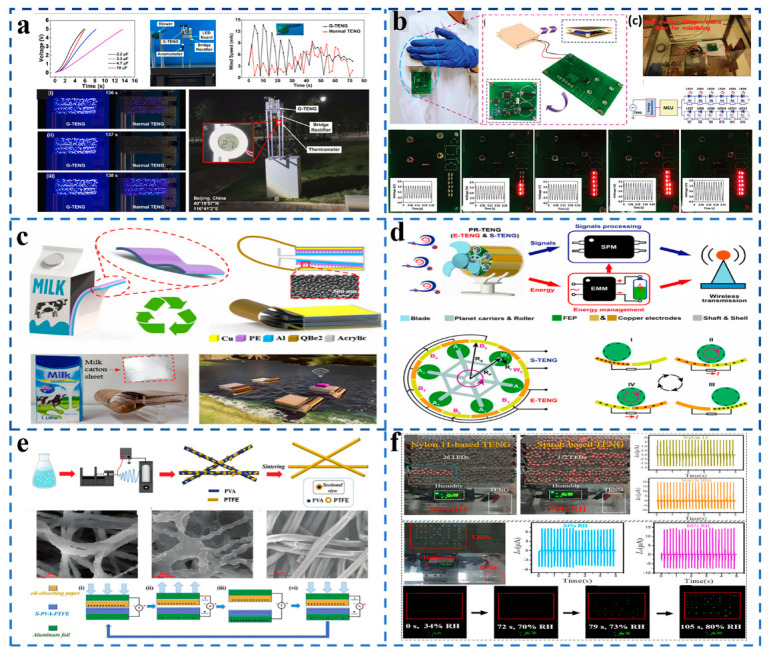
(**a**) A TENG that can stably collect wind energy. (Reproduced with permission [45]. Copyright 2021 Elsevier). (**b**) A TENG used to monitor the toxic gas of aniline. (Reproduced with permission [46]. Copyright 2020 John Wiley and Sons). (**c**) A TENG made of waste milk cartons can be used for actual environmental monitoring. (Reproduced with permission [47]. Copyright 2018 Elsevier). (**d**) A complete self-powered intelligent wireless sensor system can be realized through a kind of TENG. (Reproduced with permission [48]. Copyright 2021 American Chemical Society). (**e**) A kind of acid- and alkali-resistant TENG. (Reproduced with permission [49]. Copyright 2020 Royal Society of Chemistry). (**f**) A TENG with high output performance in a 95% humidity environment. (Reproduced with permission [50]. Copyright 2020 Elsevier).

**Figure 7 sensors-21-07129-f007:**
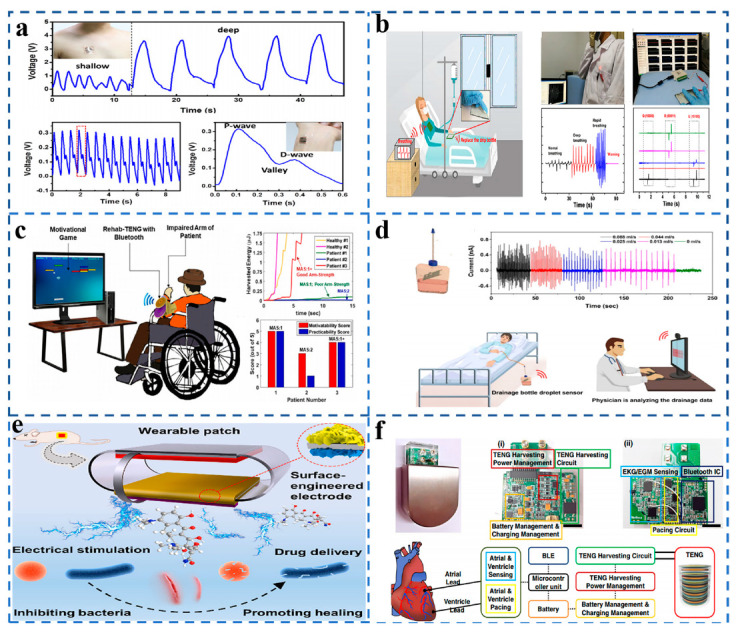
(**a**) A TENG used as an ultra-sensitive pressure sensor for breathing and pulse monitoring. (Reproduced with permission [51]. Copyright 2019 Elsevier). (**b**) A TENGTENG used as a vital sign monitoring and finger click communication sensor. (Reproduced with permission [52]. Copyright 2019 Elsevier). (**c**) Wearable TENG–based exercise system for upper limb rehabilitation post neurological injuries. (Reproduced with permission [53]. Copyright 2021 Elsevier). (**d**) Superhydrophobic liquid–solid Contact TENG as a droplet sensor for biomedical applications. (Reproduced with permission [54]. Copyright 2020 American Chemical Society). (**e**) TENG promoting the healing of infected wounds. (Reproduced with permission [55]. Copyright 2021 Elsvier). (**f**) Self-rechargeable cardiac pacemaker system with TENG. (Reproduced with permission [56]. Copyright 2021 Springer Nature).

**Figure 8 sensors-21-07129-f008:**
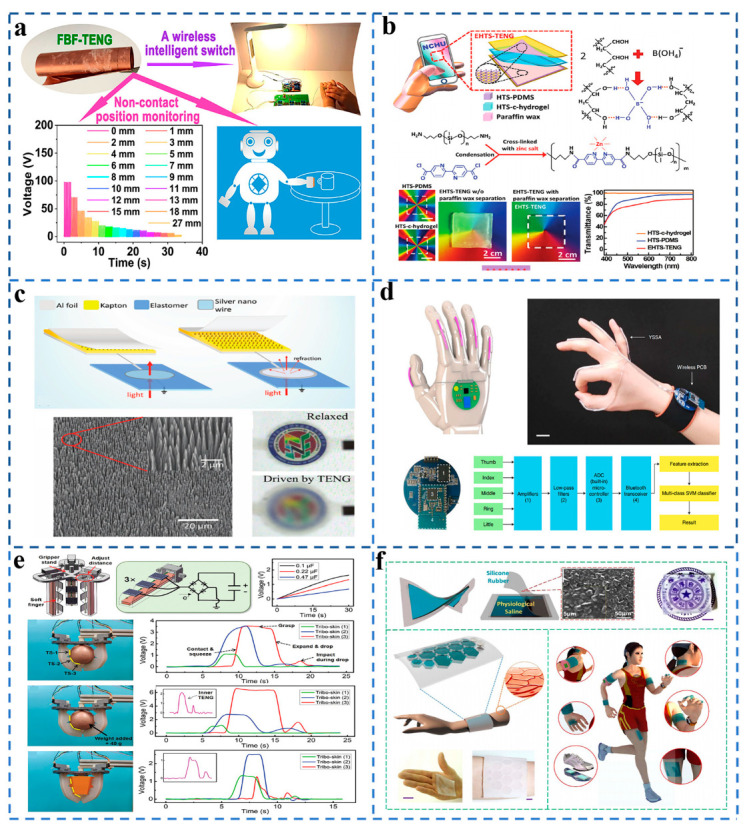
(**a**) Fish bladder film-based TENG used for smart electronic skin. (Reproduced with permission [59]. Copyright 2020 American Chemical Society). (**b**) Self-healing and self-powered electronic skin. (Reproduced with permission [60]. Copyright 2019 John Wiley and Sons). (**c**) The optical modulator realized by TENG. (Reproduced with permission [61]. Copyright 2017 John Wiley and Sons). (**d**) Intelligent sign language translation device. (Reproduced with permission [64]. Copyright 2020 Springer Nature). (**e**) Concept illustration of the fabrication process of Ag NFs electrode with bioinspired net structure. (Reproduced with permission [67]. Copyright 2020 Springer Nature). (**f**) Design of cross-type triboelectric sensor matrix for tactile imaging. (Reproduced with permission [68]. Copyright 2018 John Wiley and Sons).

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
