# Peer review of "Recent Progress in Self-Powered Sensors Based on Triboelectric Nanogenerators"

_sensors, 2021, doi:10.3390/s21217129_

Round 1

Reviewer 1 Report

The manuscript reports on the recent progress in Self-Powered Sensors Based on Triboelectric Nanogenerators. The manuscript is well-written, and the sections are systematically arranged. I would recommend publishing this manuscript after a minor revision.

Comments:
1. Please revised the title. Is that process or progress?
2. More discussion of the motivation of current work is needed in the introduction section.
3. I would suggest adding the new section "sign language interpretation" in the R&D section. Nature communications 12, no. 1 (2021): 1-13, Nano Energy 76 (2020): 105071 etc.

Author Response

Response to Reviewer 1 Comments

Point 1: Please revised the title. Is that process or progress?

Response 1: Thank you for your careful reading and suggestion. The title has been changed Recent Progress in Self-Powered Sensors Based on Triboelectric Nanogenerators.

Point 2: More discussion of the motivation of current work is needed in the introduction section.

Response 2: Thank you very much for your careful review. TENG-based self-powered sensors show strong adaptability in the field of sustainability. After nearly a decade of development, it has flourished in several fields. It could solve some of the key problems facing the world's sustainable development. This is also the goal of current and future TENG-based self-powered sensors. Therefore, the re-search progress of TENG-based self-powered sensor is summarized and prospected. We believe that TENG-based self-powered sensors will develop more rapidly in the future and will soon be industrialized.

Point 3. I would suggest adding the new section "sign language interpretation" in the R&D section. Nature communications 12, no. 1 (2021): 1-13, Nano Energy 76 (2020): 105071 etc.

Response 3: Thanks for your professional advice. TENG, as a smart skin, has irreplaceable portability and comfort. TENG - based self-powered sensor has great potential in sign language interpretation. Park et al. present a human skin inspired highly sensitive and sustainable, self-powered triboelectric flex sensor (STFS) for sensing the human finger gestures and convert the correlated sign language into the voice and text. Lee et al. showed a sign language recognition and communication system comprising triboelectric sensor integrated gloves, AI block, and the VR interaction interface.

Reviewer 2 Report

Nice overview about TENG's. Interesting for a wide range of readers. I am missing discussions with other harvesting principles, especialy polarisation changing principles (i.e. piezoelectric, pyroelectric, flexoelectric). Showing possible fictitious applications simulates a non-existent realization and marketing. Unfortunately, it is not emphasized what the few specified output powers, energies, voltages and currents refer to and under which conditions they were measured. Except for Figs. 1 and 2, the images shown are much too small and cannot be understood by the reader    

Author Response

Response to Reviewer 2 Comments

Point 1: Nice overview about TENG's. Interesting for a wide range of readers. I am missing discussions with other harvesting principles, especially polarisation changing principles (i.e. piezoelectric, pyroelectric, flexoelectric). Showing possible fictitious applications simulates a non-existent realization and marketing. Unfortunately, it is not emphasized what the few specified output powers, energies, voltages and currents refer to and under which conditions they were measured. Except for Figs. 1 and 2, the images shown are much too small and cannot be understood by the reader 

Response 1: Thanks for your professional advice.

  1. The corresponding content has been modified. We have discussed sensors with other harvesting principles (piezoelectric, pyroelectric, flexoelectric) in manuscript.
  2. We have added conditions of TENGs’ output powers, energies, voltages and currents in manuscript.
  3. We have used high resolution images in manuscript.
